# DATA-EFFICIENT REINFORCEMENT LEARNING WITH SELF-PREDICTIVE REPRESENTATIONS

**Max Schwarzer**[*]
Mila, Université de Montréal

**Ankesh Anand**[*]
Mila, Université de Montréal
Microsoft Research

**Rishab Goel**
Mila

**R Devon Hjelm**
Microsoft Research
Mila, Université de Montréal

**Aaron Courville**
Mila, Université de Montréal
CIFAR Fellow

**Philip Bachman**
Microsoft Research

## ABSTRACT

While deep reinforcement learning excels at solving tasks where large amounts of data can be collected through virtually unlimited interaction with the environment, learning from limited interaction remains a key challenge. We posit that an agent can learn more efficiently if we augment reward maximization with self-supervised objectives based on structure in its visual input and sequential interaction with the environment. Our method, Self-Predictive Representations (SPR), trains an agent to predict its own latent state representations multiple steps into the future. We compute *target* representations for future states using an encoder which is an exponential moving average of the agent's parameters and we make predictions using a learned transition model. On its own, this future prediction objective outperforms prior methods for sample-efficient deep RL from pixels. We further improve performance by adding data augmentation to the future prediction loss, which forces the agent's representations to be consistent across multiple views of an observation. Our full self-supervised objective, which combines future prediction and data augmentation, achieves a median human-normalized score of 0.415 on Atari in a setting limited to 100k steps of environment interaction, which represents a 55% relative improvement over the previous state-of-the-art. Notably, even in this limited data regime, SPR exceeds expert human scores on 7 out of 26 games. We've made the code associated with this work available at https://github.com/mila-iqia/spr.

## 1 INTRODUCTION

Deep Reinforcement Learning (deep RL, François-Lavet et al., 2018) has proven to be an indispensable tool for training successful agents on difficult sequential decision-making problems (Bellemare et al., 2013; Tassa et al., 2018). The success of deep RL is particularly noteworthy in highly complex, strategic games such as StarCraft (Vinyals et al., 2019) and DoTA2 (OpenAI et al., 2019), where deep RL agents now surpass expert human performance in some scenarios.

Deep RL involves training agents based on large neural networks using large amounts of data (Sutton, 2019), a trend evident across both model-based (Schrittwieser et al., 2020) and model-free (Badia et al., 2020) learning. The sample complexity of such state-of-the-art agents is often incredibly high: MuZero (Schrittwieser et al., 2020) and Agent-57 (Badia et al., 2020) use 10-50 years of experience per Atari game, and OpenAI Five (OpenAI et al., 2019) uses *45,000 years* of experience to accomplish its remarkable performance. This is clearly impractical: unlike easily-simulated environments such as video games, collecting interaction data for many real-world tasks is costly, making improved *data efficiency* a prerequisite for successful use of deep RL in these settings (Dulac-Arnold et al., 2019).

---

[*]Equal contribution; the order of first authors was determined by a coin flip. {schwarzm, ankesh.anand}@mila.quebec

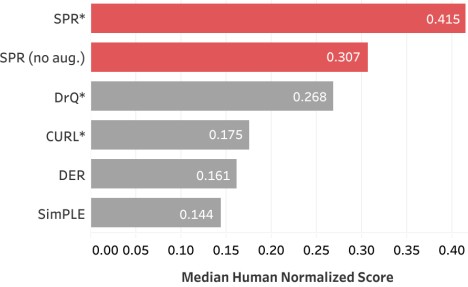 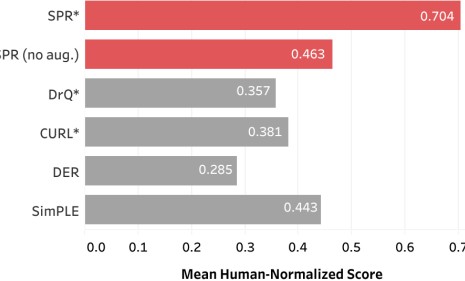

Figure 1: Median and Mean Human-Normalized scores of different methods across 26 games in the Atari 100k benchmark (Kaiser et al., 2019), averaged over 10 random seeds for SPR, and 5 seeds for most other methods except CURL, which uses 20. Each method is allowed access to only 100k environment steps or 400k frames per game. (*) indicates that the method uses data augmentation. SPR achieves state-of-art results on both mean and median human-normalized scores. Note that, even without data augmentation, SPR still outperforms all prior methods on both metrics.

Meanwhile, new self-supervised representation learning methods have significantly improved data efficiency when learning new vision and language tasks, particularly in low data regimes or semi-supervised learning (Xie et al., 2019; Hénaff et al., 2019; Chen et al., 2020b). Self-supervised methods improve data efficiency by leveraging a nearly limitless supply of training signal from tasks generated on-the-fly, based on "views" drawn from the natural structure of the data (e.g., image patches, data augmentation or temporal proximity, see Doersch et al., 2015; Oord et al., 2018; Hjelm et al., 2019; Tian et al., 2019; Bachman et al., 2019; He et al., 2020; Chen et al., 2020a).

Motivated by successes in semi-supervised and self-supervised learning (Tarvainen & Valpola, 2017; Xie et al., 2019; Grill et al., 2020), we train better state representations for RL by forcing representations to be temporally predictive and consistent when subject to data augmentation. Specifically, we extend a strong model-free agent by adding a dynamics model which predicts future latent representations provided by a parameter-wise exponential moving average of the agent itself. We also add data augmentation to the future prediction task, which enforces consistency across different views of each observation. Contrary to some methods (Kaiser et al., 2019; Hafner et al., 2019), our dynamics model operates entirely in the latent space and does not rely on reconstructing raw states.

We evaluate our method, which we call Self-Predictive Representations (SPR), on the 26 games in the Atari 100k benchmark (Kaiser et al., 2019), where agents are allowed only 100k steps of environment interaction (producing 400k frames of input) per game, which roughly corresponds to two hours of real-time experience. Notably, the human experts in Mnih et al. (2015) and Van Hasselt et al. (2016) were given the same amount of time to learn these games, so a budget of 100k steps permits a reasonable comparison in terms of data efficiency.

In our experiments, we augment a modified version of Data-Efficient Rainbow (DER) (van Hasselt et al., 2019) with the SPR loss, and evaluate versions of SPR with and without data augmentation. We find that each version is superior to controlled baselines. When coupled with data augmentation, SPR achieves a median score of 0.415, which is a state-of-the-art result on this benchmark, outperforming prior methods by a significant margin. Notably, SPR also outperforms human expert scores on 7 out of 26 games while using roughly the same amount of in-game experience.

## 2    METHOD

We consider reinforcement learning (RL) in the standard Markov Decision Process (MDP) setting where an agent interacts with its environment in *episodes*, each consisting of sequences of observations, actions and rewards. We use $s_t$, $a_t$ and $r_t$ to denote the state, the action taken by the agent and the reward received at timestep $t$. We seek to train an agent whose expected cumulative reward in each episode is maximized. To do this, we combine a strong model-free RL algorithm, Rainbow (Hessel

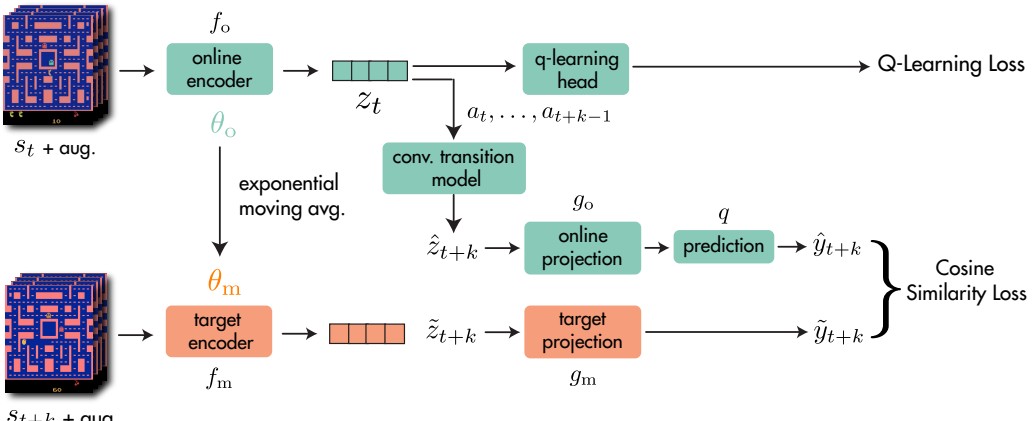

Figure 2: An illustration of the full SPR method. Representations from the online encoder are used in the reinforcement learning task and for prediction of future representations from the target encoder via the transition model. The target encoder and projection head are defined as an exponential moving average of their online counterparts and are not updated via gradient descent. For brevity, we illustrate only the $k^{\text{th}}$ step of future prediction, but in practice we compute the loss over all steps from 1 to $K$. Note: our implementation for this paper includes $g_o$ in the Q-learning head.

et al., 2018), with Self-Predictive Representations as an auxiliary loss to improve sample efficiency. We now describe our overall approach in detail.

## 2.1 DEEP Q-LEARNING

We focus on the Atari Learning Environment (Bellemare et al., 2013), a challenging setting where the agent takes discrete actions while receiving purely visual, pixel-based observations. A prominent method for solving Atari, Deep Q Networks (Mnih et al., 2015), trains a neural network $Q_\theta$ to approximate the agent's current *Q-function* (policy evaluation) while updating the agent's policy greedily with respect to this Q-function (policy improvement). This involves minimizing the error between predictions from $Q_\theta$ and a target value estimated by $Q_\xi$, an earlier version of the network:

$$\mathcal{L}_\theta^{DQN} = \Big( Q_\theta(s_t, a_t) - (r_t + \gamma \max_a Q_\xi(s_{t+1}, a)) \Big)^2. \tag{1}$$

Various improvements have been made over the original DQN: Distributional RL (Bellemare et al., 2017) models the full distribution of future reward rather than just the mean, Dueling DQN (Wang et al., 2016) decouples the *value* of a state from the *advantage* of taking a given action in that state, Double DQN (Van Hasselt et al., 2016) modifies the Q-learning update to avoid overestimation due to the max operation, among many others. Rainbow (Hessel et al., 2018) consolidates these improvements into a single combined algorithm and has been adapted to work well in data-limited regimes (van Hasselt et al., 2019).

## 2.2 SELF-PREDICTIVE REPRESENTATIONS

For our auxiliary loss, we start with the intuition that encouraging state representations to be predictive of future states given future actions should improve the data efficiency of RL algorithms. Let $(s_{t:t+K}, a_{t:t+K})$ denote a sequence of $K + 1$ previously experienced states and actions sampled from a replay buffer, where $K$ is the maximum number of steps into the future which we want to predict. Our method has four main components which we describe below:

- **Online and Target networks**: We use an *online encoder* $f_o$ to transform observed states $s_t$ into representations $z_t \triangleq f_o(s_t)$. We use these representations in an objective that encourages them to be *predictive* of future observations up to some fixed temporal offset $K$, given a sequence of $K$ actions

to perform. We augment each observation $s_t$ independently when using data augmentation. Rather than predicting representations produced by the online encoder, we follow prior work (Tarvainen & Valpola, 2017; Grill et al., 2020) by computing target representations for future states using a *target encoder* $f_{\mathrm{m}}$, whose parameters are an exponential moving average (EMA) of the online encoder parameters. Denoting the parameters of $f_{\mathrm{o}}$ as $\theta_{\mathrm{o}}$, those of $f_{\mathrm{m}}$ as $\theta_{\mathrm{m}}$, and the EMA coefficient as $\tau \in [0, 1)$, the update rule for $\theta_{\mathrm{m}}$ is:

$$\theta_{\mathrm{m}} \leftarrow \tau\theta_{\mathrm{m}} + (1 - \tau)\theta_{\mathrm{o}}. \tag{2}$$

The target encoder is not updated via gradient descent. The special case $\tau = 0$, $\theta_m = \theta_o$ is noteworthy, as it performs well when regularization is already provided by data augmentation.

- **Transition Model:** For the prediction objective, we generate a sequence of $K$ predictions $\hat{z}_{t+1:t+K}$ of future state representations $\tilde{z}_{t+1:t+K}$ using an action-conditioned *transition model* $h$. We compute $\hat{z}_{t+1:t+K}$ iteratively: $\hat{z}_{t+k+1} \triangleq h(\hat{z}_{t+k}, a_{t+k})$, starting from $\hat{z}_t \triangleq z_t \triangleq f_o(s_t)$. We compute $\tilde{z}_{t+1:t+K}$ by applying the target encoder $f_m$ to the observed future states $s_{t+1:t+K}$: $\tilde{z}_{t+k} \triangleq f_m(s_{t+k})$. The transition model and prediction loss operate in the latent space, thus avoiding pixel-based reconstruction objectives. We describe the architecture of $h$ in section 2.3.

- **Projection Heads:** We use online and target projection heads $g_o$ and $g_m$ (Chen et al., 2020a) to project online and target representations to a smaller latent space, and apply an additional *prediction head* $q$ (Grill et al., 2020) to the online projections to predict the target projections:

$$\hat{y}_{t+k} \triangleq q(g_o(\hat{z}_{t+k})), \ \forall \hat{z}_{t+k} \in \hat{z}_{t+1:t+K}; \quad \tilde{y}_{t+k} \triangleq g_m(\tilde{z}_{t+k}), \ \forall \tilde{z}_{t+k} \in \tilde{z}_{t+1:t+K}. \tag{3}$$

The target projection head parameters are given by an EMA of the online projection head parameters, using the same update as the online and target encoders.

- **Prediction Loss**: We compute the future prediction loss for SPR by summing over cosine similarities[1] between the predicted and observed representations at timesteps $t + k$ for $1 \le k \le K$:

$$\mathcal{L}_\theta^{\mathrm{SPR}}(s_{t:t+K}, a_{t:t+K}) = -\sum_{k=1}^{K} \left( \frac{\tilde{y}_{t+k}}{||\tilde{y}_{t+k}||_2} \right)^\top \left( \frac{\hat{y}_{t+k}}{||\hat{y}_{t+k}||_2} \right), \tag{4}$$

where $\tilde{y}_{t+k}$ and $\hat{y}_{t+k}$ are computed from $(s_{t:t+K}, a_{t:t+K})$ as we just described.

We call our method Self-Predictive Representations (SPR), following the predictive nature of the objective and the use of an exponential moving average target network similar to (Tarvainen & Valpola, 2017; He et al., 2020). During training, we combine the SPR loss with the Q-learning loss for Rainbow. The SPR loss affects $f_o$, $g_o$, $q$ and $h$. The Q-learning loss affects $f_o$ and the Q-learning head, which contains additional layers specific to Rainbow. Denoting the Q-learning loss from Rainbow as $\mathcal{L}_\theta^{\mathrm{RL}}$, our full optimization objective is: $\mathcal{L}_\theta^{\mathrm{total}} = \mathcal{L}_\theta^{\mathrm{RL}} + \lambda\mathcal{L}_\theta^{\mathrm{SPR}}$.

Unlike some other proposed methods for representation learning in reinforcement learning (Srinivas et al., 2020), SPR can be used with or without data augmentation, including in contexts where data augmentation is unavailable or counterproductive. Moreover, compared to related work on contrastive representation learning, SPR does not use negative samples, which may require careful design of contrastive tasks, large batch sizes (Chen et al., 2020a), or the use of a buffer to emulate large batch sizes (He et al., 2020)

## 2.3 TRANSITION MODEL ARCHITECTURE

For the transition model $h$, we apply a convolutional network directly to the $64 \times 7 \times 7$ spatial output of the convolutional encoder $f_o$. The network comprises two 64-channel convolutional layers with $3 \times 3$ filters, with batch normalization (Ioffe & Szegedy, 2015) after the first convolution and ReLU nonlinearities after each convolution. We append a one-hot vector representing the action taken to each location in the input to the first convolutional layer, similar to Schrittwieser et al. (2020). We use a maximum prediction depth of $K = 5$, and we truncate calculation of the SPR loss at episode boundaries to avoid encoding environment reset dynamics into the model.

---

[1]Cosine similarity is linearly related to the "normalized L2" loss used in BYOL (Grill et al., 2020).

---

**Algorithm 1:** Self-Predictive Representations

---

Denote parameters of online encoder $f_o$ and projection $g_o$ as $\theta_o$
Denote parameters of target encoder $f_m$ and projection $g_m$ as $\theta_{\mathrm{m}}$
Denote parameters of transition model $h$, predictor $q$ and Q-learning head as $\phi$
Denote the maximum prediction depth as $K$, batch size as $N$
initialize replay buffer $B$
**while** *Training* **do**
    collect experience $(s, a, r, s')$ with $(\theta_o, \phi)$ and add to buffer $B$
    sample a minibatch of sequences of $(s, a, r, s') \sim B$
    **for** $i$ *in range*$(0, N)$ **do**
        **if** *augmentation* **then**
            $s^i \leftarrow \mathrm{augment}(s^i); s'^i \leftarrow \mathrm{augment}(s'^i)$
        **end**
        $z_0^i \leftarrow f_\theta(s_0^i)$                        // online representations
        $l^i \leftarrow 0$
        **for** $k$ *in (1, …, K)* **do**
            $\hat{z}_k^i \leftarrow h(\hat{z}_{k-1}^i, a_{k-1}^i)$     // latent states via transition model
            $\tilde{z}_k^i \leftarrow f_m(s_k^i)$                 // target representations
            $\hat{y}_k^i \leftarrow q(g_o(\hat{z}_k^i)), \tilde{y}_k^i \leftarrow g_m(\tilde{z}_k^i)$       // projections
            $l^i \leftarrow l^i - \left(\frac{\tilde{y}_k^i}{||\tilde{y}_k^i||_2}\right)^{\top} \left(\frac{\hat{y}_k^i}{||\hat{y}_k^i||_2}\right)$       // SPR loss at step $k$
        **end**
        $l^i \leftarrow \lambda l^i + \mathrm{RL\ loss}(s^i, a^i, r^i, s'^i; \theta_o)$   // Add RL loss for batch with $\theta_o$
    **end**
    $l \leftarrow \frac{1}{N} \sum_{i=0}^{N} l^i$                   // average loss over minibatch
    $\theta_o, \phi \leftarrow \mathrm{optimize}((\theta_o, \phi), l)$         // update online parameters
    $\theta_m \leftarrow \tau\theta_o + (1 - \tau)\theta_m$         // update target parameters
**end**

---

## 2.4 DATA AUGMENTATION

When using augmentation, we use the same set of image augmentations as in DrQ from Yarats et al. (2021), consisting of small random shifts and color jitter. We normalize activations to lie in $[0, 1]$ at the output of the convolutional encoder and transition model, as in Schrittwieser et al. (2020). We use Kornia (Riba et al., 2020) for efficient GPU-based data augmentations.

When not using augmentation, we find that SPR performs better when dropout (Srivastava et al., 2014) with probability 0.5 is applied at each layer in the online and target encoders. This is consistent with Laine & Aila (2017); Tarvainen & Valpola (2017), who find that adding noise inside the network is important when not using image-specific augmentation, as proposed by Bachman et al. (2014).

## 2.5 IMPLEMENTATION DETAILS

For our Atari experiments, we largely follow van Hasselt et al. (2019) for DQN hyperparameters, with four exceptions. We follow DrQ (Yarats et al., 2021) by: using the 3-layer convolutional encoder from Mnih et al. (2015), using 10-step returns instead of 20-step returns for Q-learning, and not using a separate DQN target network when using augmentation. We also perform two gradient steps per environment step instead of one. We show results for this configuration with and without augmentation in Table 5, and confirm that these changes are not themselves responsible for our performance. We reuse the first layer of the DQN MLP head as the SPR projection head $g_o$. When using dueling DQN (Wang et al., 2016), $g_o$ concatenates the outputs of the first layers of the value and advantage heads. When these layers are noisy (Fortunato et al., 2018), $g_o$ does not use the noisy parameters. Finally, we parameterize the predictor $q$ as a linear layer. We use $\tau = 0.99$ when augmentation is disabled and $\tau = 0$ when enabled. For $\mathcal{L}_\theta^{\mathrm{total}} = \mathcal{L}_\theta^{\mathrm{RL}} + \lambda\mathcal{L}_\theta^{\mathrm{SPR}}$, we use $\lambda = 2$.

Hyperparameters were tuned over a subset of games (following Mnih et al., 2015; Machado et al., 2018). We list the complete hyperparameters in Table 3.

Our implementation uses `rlpyt` (Stooke & Abbeel, 2019) and `PyTorch` (Paszke et al., 2019). We find that SPR modestly increases the time required for training, which we discuss in more detail in Appendix D.

## 3 RELATED WORK

### 3.1 DATA-EFFICIENT RL:

A number of works have sought to improve sample efficiency in deep RL. SiMPLe (Kaiser et al., 2019) learns a pixel-level transition model for Atari to generate simulated training data, achieving strong results on several games in the 100k frame setting, at the cost of requiring several weeks for training. However, van Hasselt et al. (2019) and Kielak (2020) introduce variants of Rainbow (Hessel et al., 2018) tuned for sample efficiency, Data-Efficient Rainbow (DER) and OTRainbow, which achieve comparable or superior performance with far less computation.

In the context of continuous control, several works propose to leverage a latent-space model trained on reconstruction loss to improve sample efficiency (Hafner et al., 2019; Lee et al., 2019; Hafner et al., 2020). Most recently, DrQ (Yarats et al., 2021) and RAD (Laskin et al., 2020) have found that applying modest image augmentation can substantially improve sample efficiency in reinforcement learning, yielding better results than prior model-based methods. Data augmentation has also been found to improve generalization of reinforcement learning methods (Combes et al., 2018; Laskin et al., 2020) in multi-task and transfer settings. We show that data augmentation can be more effectively leveraged in reinforcement learning by forcing representations to be consistent between different augmented views of an observation while also predicting future latent states.

### 3.2 REPRESENTATION LEARNING IN RL:

Representation learning has a long history of use in RL – see Lesort et al. (2018). For example, CURL (Srinivas et al., 2020) proposed a combination of image augmentation and a contrastive loss to perform representation learning for RL. However, follow-up results from RAD (Laskin et al., 2020) suggest that most of the benefits of CURL come from image augmentation, not its contrastive loss.

CPC (Oord et al., 2018), CPC|Action (Guo et al., 2018), ST-DIM (Anand et al., 2019) and DRIML (Mazoure et al., 2020) propose to optimize various temporal contrastive losses in reinforcement learning environments. We perform an ablation comparing such temporal contrastive losses to our method in Section 5. Kipf et al. (2019) propose to learn object-oriented contrastive representations by training a structured transition model based on a graph neural network.

SPR bears some resemblance to DeepMDP (Gelada et al., 2019), which trains a transition model with an unnormalized L2 loss to predict representations of future states, along with a reward prediction objective. However, DeepMDP uses its online encoder to generate prediction targets rather than employing a target encoder, and is thus prone to representational collapse (sec. C.5 in Gelada et al. (2019)). To mitigate this issue, DeepMDP relies on an additional observation reconstruction objective. In contrast, our model is self-supervised, trained entirely in the latent space, and uses a normalized loss. Our ablations (sec. 5) demonstrate that using a target encoder has a large impact on our method.

SPR is also similar to PBL (Guo et al., 2020), which directly predicts representations of future states. However, PBL uses two separate target networks trained via gradient descent, whereas SPR uses a single target encoder, updated without backpropagation. Moreover, PBL studies multi-task generalization in the asymptotic limits of data, whereas SPR is concerned with single-task performance in low data regimes, using $0.01\%$ as much data as PBL. Unlike PBL, SPR additionally enforces consistency across augmentations, which empirically provides a large boost in performance.

## 4 RESULTS

We test SPR on the sample-efficient Atari setting introduced by Kaiser et al. (2019) and van Hasselt et al. (2019). In this task, only 100,000 environment steps of training data are available – equivalent to

400,000 frames, or just under two hours – compared to the typical standard of 50,000,000 environment steps, or roughly 39 days of experience. When used without data augmentation, SPR demonstrates scores comparable to the previous best result from Yarats et al. (2021). When combined with data augmentation, SPR achieves a median human-normalized score of 0.415, which is a new state-of-the-art result on this task. SPR achieves super-human performance on seven games in this data-limited setting: Boxing, Krull, Kangaroo, Road Runner, James Bond and Crazy Climber, compared to a maximum of two for any previous methods, and achieves scores higher than DrQ (the previous state-of-the-art method) on 23 out of 26 games. See Table 1 for aggregate metrics and Figure 3 for a visualization of results. A full list of scores is presented in Table 4 in the appendix. For consistency with previous works, we report human and random scores from Wang et al. (2016).

Table 1: Performance of different methods on the 26 Atari games considered by Kaiser et al. (2019) after 100k environment steps. Results are recorded at the end of training and averaged over 10 random seeds for SPR, 20 for CURL, and 5 for other methods. SPR outperforms prior methods on all aggregate metrics, and exceeds expert human performance on 7 out of 26 games.

| Metric | Random | Human | SimPLe | DER | OTRainbow | CURL | DrQ | SPR (no Aug) | SPR |
|---|---|---|---|---|---|---|---|---|---|
| Mean Human-Norm'd | 0.000 | 1.000 | 0.443 | 0.285 | 0.264 | 0.381 | 0.357 | 0.463 | **0.704** |
| Median Human-Norm'd | 0.000 | 1.000 | 0.144 | 0.161 | 0.204 | 0.175 | 0.268 | 0.307 | **0.415** |
| Mean DQN@50M-Norm'd | 0.000 | 23.382 | 0.232 | 0.239 | 0.197 | 0.325 | 0.171 | 0.336 | **0.510** |
| Median DQN@50M-Norm'd | 0.000 | 0.994 | 0.118 | 0.142 | 0.103 | 0.142 | 0.131 | 0.225 | **0.361** |
| # Games Superhuman | 0 | N/A | 2 | 2 | 1 | 2 | 2 | 5 | **7** |

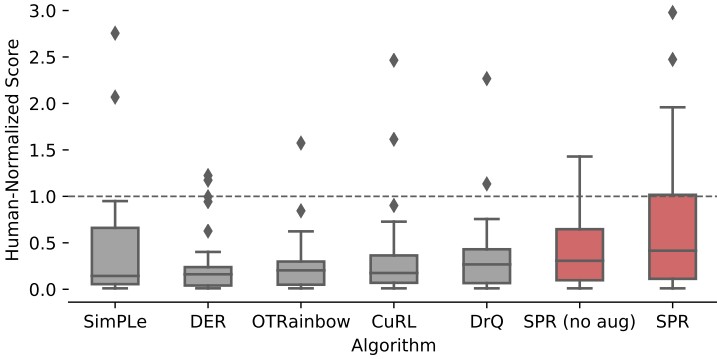

Figure 3: A boxplot of the distribution of human-normalized scores across the 26 Atari games under consideration, after 100k environment steps. The whiskers represent the interquartile range of human-normalized scores over the 26 games. Scores for each game are recorded at the end of training and averaged over 10 random seeds for SPR, 20 for CURL, and 5 for other methods.

## 4.1 EVALUATION

We evaluate the performance of different methods by computing the average episodic return at the end of training. We normalize scores with respect to expert human scores to account for different scales of scores in each game, as done in previous works. The human-normalized score of an agent on a game is calculated as $\frac{\text{agent score} - \text{random score}}{\text{human score} - \text{random score}}$ and aggregated across the 26 games by mean or median.

We find that human scores on some games are so high that differences between methods are washed out by normalization, making it hard for these games to influence aggregate metrics. Moreover, we find that the median score is typically only influenced by a handful of games. Both these factors compound together to make the median human-normalized score an unreliable metric for judging overall performance. To address this, we also report DQN-normalized scores, defined analogously to human-normalized scores and calculated using scores from DQN agents (Mnih et al., 2015) trained over 50 million steps, and report both mean and median of those metrics in all results and ablations, and plot the distribution of scores over all the games in Figure 3.

Table 2: Scores on the 26 Atari games under consideration for ablated variants of SPR. All variants listed here use data augmentation.

| Variant | Human-Normalized Score | | DQN@50M-Normalized Score | |
| --- | --- | --- | --- | --- |
| | mean | median | mean | median |
| SPR | **0.704** | **0.415** | **0.510** | **0.361** |
| 1-step SPR | 0.570 | 0.301 | 0.337 | 0.346 |
| Non-temporal SPR | 0.507 | 0.271 | 0.326 | 0.295 |
| Quadratic SPR | 0.047 | 0.040 | -0.016 | 0.031 |
| SPR without projections | 0.437 | 0.171 | 0.247 | 0.174 |
| Rainbow (controlled, w/ aug.) | 0.480 | 0.346 | 0.284 | 0.278 |

Additionally, we note that the standard evaluation protocol of evaluating over only 500,000 frames per game is problematic, as the quantity we are trying to measure is expected return over episodes. Because episodes may last up to up to 108,000 frames, this method may collect as few as four complete episodes. As variance of results is already a concern in deep RL (see Henderson et al., 2018), we recommend evaluating over 100 episodes irrespective of their length. Moreover, to address findings from Henderson et al. (2018) that comparisons based on small numbers of random seeds are unreliable, we average our results over ten random seeds, twice as many as most previous works.

## 5 ANALYSIS

**The target encoder** We find that using a separate target encoder is vital in all cases. A variant of SPR in which target representations are generated by the online encoder without a stopgradient (as done by e.g., Gelada et al., 2019) exhibits catastrophically reduced performance, with median human-normalized score of $0.278$ with augmentation versus $0.415$ for SPR. However, there is more flexibility in the EMA constant used for the target encoder. When using augmentation, a value of $\tau = 0$ performs best, while without augmentation we use $\tau = 0.99$. The success of $\tau = 0$ is interesting, since the related method BYOL reports very poor representation learning performance in this case. We hypothesize that optimizing a reinforcement learning objective in parallel with the SPR loss explains this difference, as it provides an additional gradient which discourages representational collapse. Full results for these experiments are presented in Appendix C.

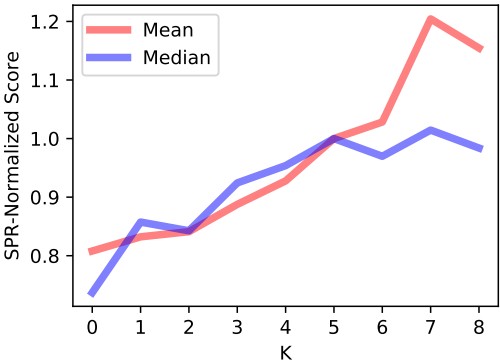

Figure 4: Performance of SPR with various prediction depths. Results are averaged across ten seeds per game, for all 26 games. To equalize the importance of games, we calculate an SPR-normalized score analogously to human-normalized scores, and show its mean and median across all 26 games. All other hyperparameters are identical to those used for SPR with augmentation.

**Dynamics modeling is key** A key distinction between SPR and other recent approaches leveraging representation learning for reinforcement learning, such as CURL (Srinivas et al., 2020) and DRIML (Mazoure et al., 2020), is our use of an explicit multi-step dynamics model. To illustrate the impact of dynamics modeling, we test SPR with a variety of prediction depths $K$. Two of these ablations, one with no dynamics modeling and one that models only a single step of dynamics, are presented in Table 2 (as *Non-temporal SPR* and *1-step SPR*), and all are visualized in Figure 4. We find that extended dynamics modeling consistently improves performance up to roughly $K = 5$. Moving beyond this continues to improve performance on a subset of games, at the cost of increased computation. Note that the non-temporal ablation we test is similar to

using BYOL (Grill et al., 2020) as an auxiliary task, with particular architecture choices made for the projection layer and predictor.

**Comparison with contrastive losses**   Though many recent works in representation learning employ contrastive learning, we find that SPR consistently outperforms both temporal and non-temporal variants of contrastive losses (see Table 6, appendix), including CURL (Srinivas et al., 2020).

**Using a quadratic loss causes collapse**   SPR's use of a cosine similarity objective (or a normalized L2 loss) sets it in contrast to some previous works, such as DeepMDP (Gelada et al., 2019), which have learned latent dynamics models by minimizing an un-normalized L2 loss over predictions of future latents. To examine the importance of this objective, we test a variant of SPR that minimizes un-normalized L2 loss (*Quadratic SPR* in Table 2), and find that it performs only slightly better than random. This is consistent with results from Gelada et al. (2019), who find that DeepMDP's representations are prone to collapse, and use an auxiliary reconstruction objective to prevent this.

**Projections are critical**   Another distinguishing feature of SPR is its use of projection and prediction networks. We test a variant of SPR that uses neither, instead computing the SPR loss directly over the $64 \times 7 \times 7$ convolutional feature map used by the transition model (*SPR without projections* in Table 2). We find that this variant has inferior performance, and suggest two possible explanations. First, the convolutional network represents only a small fraction of the capacity of SPR's network, containing only some 80,000 parameters out of a total of three to four million. Employing the first layer of the DQN head as a projection thus allows the SPR objective to affect far more of the network, while in this variant its impact is limited. Second, the effects of SPR in forcing invariance to augmentation may be undesirable at this level; as the convolutional feature map is the product of only three layers, it may be challenging to learn features that are simultaneously rich and invariant.

## 6   FUTURE WORK

Recent work in both visual (Chen et al., 2020b) and language representation learning (Brown et al., 2020) has suggested that self-supervised models trained on large datasets perform exceedingly well on downstream problems with limited data, often outperforming methods trained using only task-specific data. Future works could similarly exploit large corpora of unlabelled data, perhaps from multiple MDPs or raw videos, to further improve the performance of RL methods in low-data regimes. As the SPR objective is unsupervised, it could be directly applied in such settings.

Another interesting direction is to use the transition model learned by SPR for planning. MuZero (Schrittwieser et al., 2020) has demonstrated that planning with a model supervised via reward and value prediction can work extremely well given sufficient (massive) amounts of data. It remains unclear whether such models can work well in low-data regimes, and whether augmenting such models with self-supervised objectives such as SPR can improve their data efficiency.

It would also be interesting to examine whether self-supervised methods like SPR can improve generalization to unseen tasks or changes in environment, similar to how unsupervised pretraining on ImageNet can generalize to other datasets (He et al., 2020; Grill et al., 2020).

## 7   CONCLUSION

In this paper we introduced Self-Predictive Representations (SPR), a self-supervised representation learning algorithm designed to improve the data efficiency of deep reinforcement learning agents. SPR learns representations that are both temporally predictive and consistent across different views of environment observations, by directly predicting representations of future states produced by a target encoder. SPR achieves state-of-the-art performance on the 100k steps Atari benchmark, demonstrating significant improvements over prior work. Our experiments show that SPR is highly robust, and is able to outperform the previous state of the art when either data augmentation or temporal prediction is disabled. We identify important directions for future work, and hope continued research at the intersection of self-supervised learning and reinforcement learning leads to algorithms which rival the efficiency and robustness of humans.

ACKNOWLEDGEMENTS

We are grateful for the collaborative research environment provided by Mila and Microsoft Research. We would also like to acknowledge Hitachi for providing funding support for this project. We thank Nitarshan Rajkumar and Evan Racah for providing feeback on an earlier draft; Denis Yarats and Aravind Srinivas for answering questions about DrQ and CURL; Michal Valko, Sherjil Ozair and the BYOL team for long discussions about BYOL, and Phong Nguyen for helpful discussions. Finally, we thank Compute Canada and Microsoft Research for providing computational resources used in this project.

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

# A  ATARI DETAILS

We provide a full set of hyperparameters used in both the augmentation and no-augmentation cases in Table 3, including new hyperparameters for SPR.

Table 3: Hyperparameters for SPR on Atari, with and without augmentation.

| Parameter | Setting (for both variations) |
|---|---|
| Gray-scaling | True |
| Observation down-sampling | 84x84 |
| Frames stacked | 4 |
| Action repetitions | 4 |
| Reward clipping | [-1, 1] |
| Terminal on loss of life | True |
| Max frames per episode | 108K |
| Update | Distributional Q |
| Dueling | True |
| Support of Q-distribution | 51 |
| Discount factor | 0.99 |
| Minibatch size | 32 |
| Optimizer | Adam |
| Optimizer: learning rate | 0.0001 |
| Optimizer: $\beta_1$ | 0.9 |
| Optimizer: $\beta_2$ | 0.999 |
| Optimizer: $\epsilon$ | 0.00015 |
| Max gradient norm | 10 |
| Priority exponent | 0.5 |
| Priority correction | $0.4 \to 1$ |
| Exploration | Noisy nets |
| Noisy nets parameter | 0.5 |
| Training steps | 100K |
| Evaluation trajectories | 100 |
| Min replay size for sampling | 2000 |
| Replay period every | 1 step |
| Updates per step | 2 |
| Multi-step return length | 10 |
| Q network: channels | 32, 64, 64 |
| Q network: filter size | $8 \times 8, 4 \times 4, 3 \times 3$ |
| Q network: stride | 4, 2, 1 |
| Q network: hidden units | 256 |
| Non-linearity | ReLU |
| Target network: update period | 1 |
| $\lambda$ (SPR loss coefficient | 2 |
| $K$ (Prediction Depth) | 5 |

| Parameter | With Augmentation | Without Augmentation |
|---|---|---|
| Data Augmentation | Random shifts ($\pm 4$ pixels) & Intensity(scale=0.05) | None |
| Dropout | 0 | 0.5 |
| $\tau$ (EMA coefficient) | 0 | 0.99 |

## A.1  FULL RESULTS

We provide full results across all 26 games for the methods considered, including SPR with and without augmentation, in Table 4. Methods are ordered in rough order of their date of release or publication.

Table 4: Mean episodic returns on the 26 Atari games considered by Kaiser et al. (2019) after 100k environment steps. The results are recorded at the end of training and averaged over 10 random seeds. SPR outperforms prior methods on all aggregate metrics, and exceeds expert human performance on 7 out of 26 games while using a similar amount of experience.

| Game | Random | Human | SimPLe | DER | OTRainbow | CURL | DrQ | SPR (no Aug) | SPR |
|---|---|---|---|---|---|---|---|---|---|
| Alien | 227.8 | 7127.7 | 616.9 | 739.9 | 824.7 | 558.2 | 771.2 | **847.2** | 801.5 |
| Amidar | 5.8 | 1719.5 | 88.0 | **188.6** | 82.8 | 142.1 | 102.8 | 142.7 | 176.3 |
| Assault | 222.4 | 742.0 | 527.2 | 431.2 | 351.9 | 600.6 | 452.4 | **665.0** | 571.0 |
| Asterix | 210.0 | 8503.3 | **1128.3** | 470.8 | 628.5 | 734.5 | 603.5 | 820.2 | 977.8 |
| Bank Heist | 14.2 | 753.1 | 34.2 | 51.0 | 182.1 | 131.6 | 168.9 | **425.6** | 380.9 |
| BattleZone | 2360.0 | 37187.5 | 5184.4 | 10124.6 | 4060.6 | 14870.0 | 12954.0 | 10738.0 | **16651.0** |
| Boxing | 0.1 | 12.1 | 9.1 | 0.2 | 2.5 | 1.2 | 6.0 | 12.7 | **35.8** |
| Breakout | 1.7 | 30.5 | 16.4 | 1.9 | 9.8 | 4.9 | 16.1 | 12.9 | **17.1** |
| ChopperCommand | 811.0 | 7387.8 | **1246.9** | 861.8 | 1033.3 | 1058.5 | 780.3 | 667.3 | 974.8 |
| Crazy Climber | 10780.5 | 35829.4 | **62583.6** | 16185.3 | 21327.8 | 12146.5 | 20516.5 | 43391.0 | 42923.6 |
| Demon Attack | 152.1 | 1971.0 | 208.1 | 508.0 | 711.8 | 817.6 | **1113.4** | 370.1 | 545.2 |
| Freeway | 0.0 | 29.6 | 20.3 | **27.9** | 25.0 | 26.7 | 9.8 | 16.1 | 24.4 |
| Frostbite | 65.2 | 4334.7 | 254.7 | 866.8 | 231.6 | 1181.3 | 331.1 | 1657.4 | **1821.5** |
| Gopher | 257.6 | 2412.5 | 771.0 | 349.5 | 349.5 | 669.3 | 636.3 | 774.5 | 715.2 |
| Hero | 1027.0 | 30826.4 | 2656.6 | 6857.0 | 6458.8 | 6279.3 | 3736.3 | 5707.4 | **7019.2** |
| Jamesbond | 29.0 | 302.8 | 125.3 | 301.6 | 112.3 | **471.0** | 236.0 | 367.2 | 365.4 |
| Kangaroo | 52.0 | 3035.0 | 323.1 | 779.3 | 605.4 | 872.5 | 940.6 | 1359.5 | **3276.4** |
| Krull | 1598.0 | 2665.5 | **4539.9** | 2851.5 | 3277.9 | 4229.6 | 4018.1 | 3123.1 | 3688.9 |
| Kung Fu Master | 258.5 | 22736.3 | **17257.2** | 14346.1 | 5722.2 | 14307.8 | 9111.0 | 15469.7 | 13192.7 |
| Ms Pacman | 307.3 | 6951.6 | **1480.0** | 1204.1 | 941.9 | 1465.5 | 960.5 | 1247.7 | 1313.2 |
| Pong | -20.7 | 14.6 | **12.8** | -19.3 | 1.3 | -16.5 | -16.0 | -16.0 | -5.9 |
| Private Eye | 24.9 | 69571.3 | 58.3 | 97.8 | 100.0 | **218.4** | -13.6 | 52.6 | 124.0 |
| Qbert | 163.9 | 13455.0 | **1288.8** | 1152.9 | 509.3 | 1042.4 | 854.4 | 606.6 | 669.1 |
| Road Runner | 11.5 | 7845.0 | 5640.6 | 9600.0 | 2696.7 | 5661.0 | 8895.1 | 10511.0 | **14220.5** |
| Seaquest | 68.4 | 42054.7 | **683.3** | 354.1 | 286.9 | 384.5 | 301.2 | 580.8 | 583.1 |
| Up N Down | 533.4 | 11693.2 | 3350.3 | 2877.4 | 2847.6 | 2955.2 | 3180.8 | 6604.6 | **28138.5** |
| Mean Human-Norm'd | 0.000 | 1.000 | 0.443 | 0.285 | 0.264 | 0.381 | 0.357 | 0.463 | **0.704** |
| Median Human-Norm'd | 0.000 | 1.000 | 0.144 | 0.161 | 0.204 | 0.175 | 0.268 | 0.307 | **0.415** |
| Mean DQN@50M-Norm'd | 0.000 | 23.382 | 0.232 | 0.239 | 0.197 | 0.325 | 0.171 | 0.336 | **0.510** |
| Median DQN@50M-Norm'd | 0.000 | 0.994 | 0.118 | 0.142 | 0.103 | 0.142 | 0.131 | 0.225 | **0.361** |
| # Superhuman | 0 | N/A | 2 | 2 | 1 | 2 | 2 | 5 | **7** |

Table 5: Scores on the 26 Atari games under consideration for our controlled Rainbow implementation with and without augmentation, compared to previous methods. The high mean DQN-normalized score of our DQN without augmentation is due to an atypically high score on Private Eye, a hard exploration game on which the original DQN achieves a low score.

| Variant | Human-Normalized Score | | DQN@50M-Normalized Score | |
|---|---|---|---|---|
| | mean | median | mean | median |
| Rainbow (controlled, no aug) | 0.240 | 0.204 | 0.374 | 0.149 |
| OTRainbow | 0.264 | 0.204 | 0.197 | 0.103 |
| DER | 0.285 | 0.161 | 0.239 | 0.142 |
| Rainbow (controlled, w/ aug) | 0.480 | 0.346 | 0.284 | 0.278 |
| DrQ | 0.357 | 0.268 | 0.171 | 0.131 |

## A.2  CONTROLLED BASELINES

To ensure that the minor hyper-parameter changes we make to the DER baseline are not solely responsible for our improved performance, we perform controlled experiments using the same hyper-parameters and same random seeds for baselines. We find that our controlled Rainbow implementation without augmentation is slightly stronger than Data-Efficient Rainbow but comparable to Overtrained Rainbow (Kielak, 2020), while with augmentation enabled our results are somewhat stronger than DrQ.[2] None of these methods, however, are close to the performance of SPR.

---

[2]This is perhaps not surprising, given that the model used by DrQ omits many of the components of Rainbow.

Table 6: Scores on the 26 Atari games under consideration for various contrastive alternatives to SPR implemented in our codebase. All variants listed here use data augmentation.

| Variant | Human-Normalized Score | | DQN@50M-Normalized Score | |
|---|---|---|---|---|
| | mean | median | mean | median |
| SPR | 0.704 | 0.415 | 0.510 | 0.361 |
| Rainbow (controlled) | 0.480 | 0.346 | 0.284 | 0.278 |
| Non-temporal contrastive | 0.379 | 0.200 | 0.268 | 0.179 |
| 1-step contrastive | 0.473 | 0.231 | 0.280 | 0.213 |
| 5-step contrastive | 0.506 | 0.172 | 0.239 | 0.142 |
| Uniformity loss | 0.422 | 0.176 | 0.271 | 0.144 |

## B    COMPARISON WITH A CONTRASTIVE LOSS

To compare SPR with alternative methods drawn from contrastive learning, we examine several variants of a contrastive losses based on InfoNCE (Oord et al., 2018):

- A contrastive loss based solely on different views of the same state, similar to CURL (Srinivas et al., 2020).

- A temporal contrastive loss with both augmentation and where targets are drawn one step in the future, equivalent to single-step CPC (Oord et al., 2018).

- A temporal contrastive loss with an explicit dynamics model, similar to CPC|Action (Guo et al., 2018). Predictions are made up to five steps in the future, and encodings of every state except $s_{t+k}$ are used as negative samples for $s_{t+k}$.

- A soft contrastive approach inspired by Wang & Isola (2020), who propose to decouple the repulsive and attractive effects of contrastive learning into two separate losses, one of which is similar to the SPR objective and encourages representations to be invariant to augmentation or noise, and one of which encourages representations to be uniformly distributed on the unit hypersphere. We optimize this uniformity objective jointly with the SPR loss, which takes the role of the "invariance" objective proposed by (Wang & Isola, 2020). We use $t = 2$ in the uniformity loss, and give it a weight equal to that given to the SPR loss, based on hyperparameters used by Wang & Isola (2020).

To create as fair a comparison as possible, we use the same augmentation (random shifts and intensity) and the same Rainbow hyperparameters as in SPR with augmentation. As in SPR, we calculate contrastive losses using the output of the first layer of the Q-head MLP, with a bilinear classifier (as in Oord et al., 2018). Following Chen et al. (2020a), we use annealed cosine similarities with a temperature of 0.1 in the contrastive loss. We present results in Table 6.

Although all of these variants outperform the previous contrastive result on this task, CURL, none of them substantially improve performance over the controlled Rainbow they use as a baseline. We consider these results broadly consistent with those of CURL, which observes a relatively small performance boost over their baseline, Data-Efficient Rainbow (van Hasselt et al., 2019).

## C    THE ROLE OF THE TARGET ENCODER IN SPR

We consider several variants of SPR with the target network modified, and present aggregate metrics for these experiments in Table 7. We first evaluate a a variant of SPR in which target representations are drawn from the online encoder and gradients allowed to propagate into the online encoder through them, effectively allowing the encoder to learn to make its representations more predictable. We find that this leads to drastic reductions in performance both with and without augmentation, which we attribute to representational collapse.

Table 7: Scores on the 26 Atari games under consideration for variants of SPR with different target encoder schemes, without augmentation.

| Variant Without Augmentation | Human-Normalized Score mean | median | DQN@50M-Normalized Score mean | median |
|---|---|---|---|---|
| SPR | 0.463 | 0.307 | 0.336 | 0.225 |
| No Stopgradient | 0.375 | 0.208 | 0.301 | 0.233 |
| With Augmentation | | | | |
| SPR | 0.704 | 0.415 | 0.510 | 0.361 |
| No Stopgradient | 0.515 | 0.278 | 0.344 | 0.231 |

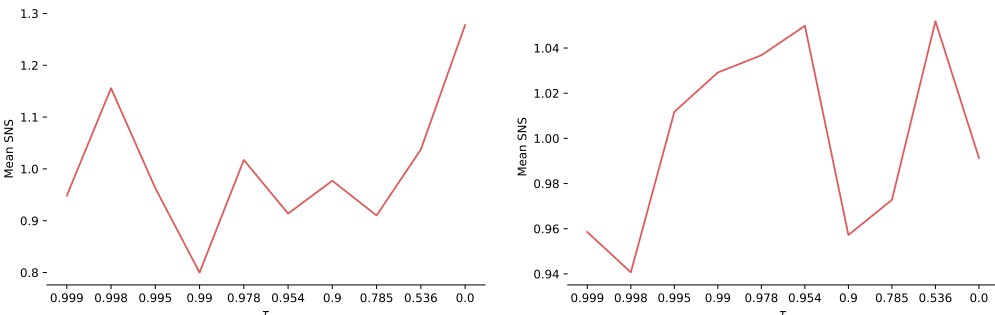

Figure 5: Performance on a subset of 10 Atari games for different values of the EMA parameter $\tau$ with augmentation (left) and without (right). Scores are averaged across 10 seeds per game for each value of $\tau$. Self-normalized score is calculated separately for the augmentation and no-augmentation cases.

To illustrate the influence of the EMA constant $\tau$, we evaluate $\tau$ at 9 values logarithmically interpolated[3] between 0.999 and 0 on a subset of 10 Atari games.[4] We use 10 seeds per game, and evaluate SPR both with and without augmentation; parameters other than $\tau$ are identical to those listed in Table 3. To equalize the importance of games in this analysis, we normalize by the average score across all tested values of $\tau$ for each game to calculate a *self-normalized* score, as $\text{score}_{\text{sns}} \triangleq \frac{\text{agent score} - \text{random score}}{\text{average score} - \text{random score}}$.

We test $SPR$ both with and without augmentation, and calculate the self-normalized score separately between these cases. Results are shown in Figure 5. With augmentation, we observe a clear peak in performance at $\tau = 0$, equivalent to a target encoder with no EMA-based smoothing. Without augmentation, however, the story is less clear, and the method appears less sensitive to $\tau$ (note y-axis scales). We use $\tau = 0.99$ in this case, based on its reasonable performance and consistency with prior work (e.g., Grill et al., 2020). Overall, however, we note that SPR does not appear overly sensitive to $\tau$, unlike purely unsupervised methods such as BYOL; in no case does SPR fail to train.

We hypothesize that the difference between the augmentation and no-augmentation cases is partially due to augmentation rendering the stabilizing effect of using an EMA target network (e.g., as observed by Grill et al., 2020; Tarvainen & Valpola, 2017) redundant. Prior work has already noted that using an EMA target network can slow down learning early in training (Tarvainen & Valpola, 2017); in our context, where a limited number of environment samples are taken in parallel with optimization, this may "waste" environment samples by collecting them with an inferior policy. To resolve this, Tarvainen & Valpola (2017) proposed to increase $\tau$ over the course of training, slowing down changes to the target network later in training. It is possible that doing so here could allow SPR to achieve the best of both worlds, but it would require tuning an additional hyperparameter, the schedule by which $\tau$ is increased, and we thus leave this topic for future work.

---

[3]$\tau \in \{0.999, 0.9976, 0.9944, 0.9867, 0.9684, 0.925, 0.8222, 0.5783, 0\}$.

[4]Pong, Breakout, Up N Down, Kangaroo, Bank Heist, Assault, Boxing, BattleZone, Frostbite and Crazy Climber

## D   WALL CLOCK TIMES

We report wall-clock runtimes for a selection of methods in Table 8. SPR with augmentation for a 100K steps on Atari takes around 4 and a half to finish a complete training and evaluation run on a single game. We find that using data augmentation adds an overhead, and SPR without augmentation can run in just 3 hours.

SPR's wall-clock run-time compares very favorably to previous works such as SimPLe (Kaiser et al., 2019), which requires roughly three weeks to train on a GPU comparable to those used for SPR.

Table 8: Wall-clock runtimes for various algorithms for a complete training and evaluation run on a single Atari game using a P100 GPU. Rainbow (controlled) is roughly comparable to DrQ, although its runtime will differ due different DQN hyperparameters. Runtime for SimPLe is taken from its v3 version on Arxiv, although the latest version doesn't mention runtime. All runtimes are approximate, as exact running times vary from game to game.

| Model | Runtime in hours (100k env steps) |
|---|---|
| SPR | 4.6 |
| Rainbow (controlled) | 2.1 |
| SPR (No aug) | 3.0 |
| Rainbow (controlled, no aug) | 1.4 |
| SimPLe | 500 |

