# OpenReview forum: "Data-Efficient Reinforcement Learning with Self-Predictive Representations"
_ICLR.cc/2021/Conference — ICLR 2021 Spotlight_

### Official Review · AnonReviewer1 · 2020-10-28
**An interesting integration of RL and self-supervised representation learning that warrants further investigation.**

**Rating:** 6
**Confidence:** 5

**Review:**

The authors propose learning self-supervised representations that are consistent across future time steps.  To do this, the authors augment a modified Data Efficient Rainbow with the self-supervised architecture from BYOL; however, instead of applying it to two augmentations of the same view, they match a sequence of states to each other.  The online sequence is generated as follows: the current state at time t is passed through the online encoder to generate z_t, from which representations {z_t+1,...,z_t+k} are generated via an action-conditioned transition model.  These k states are then matched with the target encoding of the observations {o_t+1,...,o_t+k} via cosine similarity.  They experimented with applying augmentations, as well as dropout of the features of both online and target networks (without augmentation).  The reward is also predicted from the state z_t, to ensure that the representation learned is still useful for the task.  The authors evaluate SPR on sample-efficient Atari, which is limited to only 100k environment steps (400k frames).

The paper admirably integrates a powerful self-supervised representation learning technique with the framework of reinforcement learning, by enabling the contrasting of multiple consecutive states via the use of a multi-step dynamics model.  However, inherent to the process of adapting these techniques is the need to explore design decisions and assumptions made, and their effects.

Here are a few questions to consider, that the reader would potentially be interested in:
1) Why was the original state not also matched against a target encoding (with a different augmentation) - essentially 0-step BYOL?
2) To improve the quality of the dynamically generated states as well as the transition model itself, would it not be better to learn from the multi-step reward signal as well?   This would ensure consistency between the predicted states and the encoded states.  The gradient can be stopped before it flows through the online encoder.
3) In BYOL, the authors match one augmentation passed through the online network to another augmentation passed through the target network.  By symmetry, however, the second augmentation passed through the online network should also match the first augmentation passed through the target.  Thus, BYOL is computed as a sum of these two cosine similarity losses.  Would it not also make sense for SPR to apply this symmetry?  The representation from the past should match that of the future, but is it not desirable for future representations to match the past for consistency (within a reasonable frame length)?
4) The authors of BYOL report sensitivity to the transformation set used and the batch size, despite not relying on negatives.  Do these considerations transfer to SPR?
5) Representations learned from contrastive self-supervised techniques such as BYOL and SimCLR have been shown to be good for a variety of applications, such as linear evaluation, semi-supervised tasks, and transfer learning.  Do these benefits also apply when adapted to RL via SPR - for example, transferring across tasks?

Although there are other benchmarks that could be used to evaluate SPR (such as the DeepMind Control Suite) and present a stronger case, the existing experimental results are already compelling.  Ultimately, the idea is presented clearly and the writing is solid.  However, a more careful analysis of the implications of the design decisions made and how the pros and cons of these self-supervised representation learning approaches transfer when integrated into RL would separate this paper from an incremental work.

---

> ### Author Response · Authors · 2020-11-12
> **Response to AnonReviewer1**
>
> Thanks, these are very interesting questions, some of which we pondered over ourselves earlier in the project!  In brief:
> 1. In practice, we found that matching the initial state to a target encoding did not improve performance. Moreover, when not using augmentation, calculating the SPR objective at t=0 is quite different from BYOL, as the inputs on both sides of the prediction are identical. Since part of our goal with SPR is to design a method which works with or without augmentation, we removed the t=0 loss. Though, we do perform an ablation where we calculate the loss only at t=0 (“Non-temporal SPR” in Table 2). If you’d like, we can present ablations in the final version examining the relevance of this choice.
> 2. We found that training the transition model to predict future rewards had negligible impact on performance, and thus omitted it from the final formulation of SPR to simplify the method. We think the reason for this is the sparsity of the reward signal in many Atari games, which makes rewards a relatively weak form of supervision compared to future abstract state prediction. Moreover, since the DQN objective will already force representations to encode information about expected future rewards, this information will indirectly be learned by the transition model via the SPR objective. Though, one direction we’re considering for future work is incorporating MuZero or Dyna-Q-style use of the transition model, in which case reward prediction will be important.
> 3. Using both augmentations for both the online and target network is a little bit tricky due to the semi-supervised nature of SPR -- which augmentation would you use to calculate the RL loss? One could always use both augmentations in the RL loss, averaging their predictions. However, DrQ used a scheme similar to this in DM Control but not in Atari, leading us to suspect that it may not be beneficial in Atari compared to its computational cost. Moreover, since the augmentations we use (see point 4) are much weaker than in BYOL, we suspect that forcing this symmetry might not be as important as in BYOL, where augmentation is likely a substantial source of variance in optimization.
>
> > The representation from the past should match that of the future, but is it not desirable for future representations to match the past for consistency (within a reasonable frame length)
>
> We considered this idea, but ended up not pursuing it. The most direct symmetric solution would require learning an explicit reverse-time transition model, which would add substantial complexity to our method. Additionally, we had qualms about this method due to potential cheating by the reverse transition model. Since the encoder in Atari typically conditions on several previous states as well as the current one, a reverse transition model might be able to cheat over short timescales by peeking into these frame stacks. One could address this by increasing the reverse prediction horizon, but this would also increase compute costs. Alternatively, one could simply not provide the previous states as input to the encoder when it is being used for the reverse transition model, but this breaks the clean symmetry of the method and causes some distribution shift which might prove problematic.
>
> Having said that, backward models are a promising research direction that have found more attention lately (Xu et. al (2019), Chelu et. al (2020)), and we hope future work is able to scale to domains like Atari.
>
> [1] Xu, Danfei, et al. "Regression Planning Networks." Advances in Neural Information Processing Systems. 2019. https://papers.nips.cc/paper/2019/file/3a835d3215755c435ef4fe9965a3f2a0-Paper.pdf
>
> [2] Chelu, Veronica, Doina Precup, and Hado P. van Hasselt. "Forethought and Hindsight in Credit Assignment." Advances in Neural Information Processing Systems 33 (2020). https://papers.nips.cc/paper/2020/file/18064d61b6f93dab8681a460779b8429-Paper.pdf

---

> > ### Author Response · Authors · 2020-11-12
> > **Response to AnonReviwer1 (cont.)**
> >
> > 4. In practice, augmentation in reinforcement learning almost always consists of extremely mild transformations, such as small shifts or tiny changes in image intensity.   We use the same set of augmentations as DrQ (Kostrikov et al, 2020) which found that augmentations that are commonly used in self-supervised learning for ImageNet, such as crops, are harmful in RL.
> >
> > Making stronger augmentation useful in reinforcement learning would be an excellent avenue for future work. One possible solution for this problem might be to separate reinforcement learning and representation learning, by doing one pass through the encoder using weak augmentation for the RL loss and another pass with stronger augmentation for the representation learning loss.  However, doing so adds complexity and extra computation, so we did not pursue it for SPR.
> >
> > As for batch size, Data-Efficient Rainbow (van Hasselt et al, 2019, Figure 5) found that performing more updates with smaller minibatches was substantially better than using fewer, larger minibatches.  Again, this is a case where the combination of an RL algorithm with a representation learning algorithm adds some complexity to analyses.  However, given the small scale of the networks used by SPR, it may not be surprising that optimization at small batch sizes is relatively more stable than in BYOL, which uses large ResNets.
> >
> > 5. This is a very interesting question!   We suspect that the answer is yes, and we’re actively exploring how to use SPR for pretraining or transfer, especially in the context where rewards aren’t available during pretraining.  Giving this problem proper treatment means covering quite a lot of ground, though, so we leave this investigation for future work.
> >
> > [3] Kostrikov, Ilya, Denis Yarats, and Rob Fergus. "Image augmentation is all you need: Regularizing deep reinforcement learning from pixels." arXiv preprint arXiv:2004.13649 (2020).

---

### Official Review · AnonReviewer4 · 2020-10-28
**Official Blind Review #4**

**Rating:** 7
**Confidence:** 4

**Review:**

The paper introduces a method to improve sample-efficiency in reinforcement learning, called Self-Predictive Representations. The authors use predictions in the latent space by using a learned transition model. To further improve their method, they add data-augmentation to enforce consistent the representation to be consistent over multiple views of an observation. Their method is tested on the 100k steps Atari environment and compared to other state-of-the-art methods. The results show the proposed method to significantly outperform previously suggested methods.

In its current form, the paper is marginally below the threshold in my opinion.

The reasons for my decision are as follows:
-	The main point of contention is the choice to use different numbers for random seeding. While I understand the motivation stated by the authors to make results comparable to the original baselines/state of the art, it results in several problems. The error bars in Figure 3 are not meaningful as they are based on different numbers of seeds for different methods. Furthermore, I am surprised that the authors did cite Henderson et al., 2018, but do not address the problems that arise from using only a small number of random seeds (as shown in Henderson et al., 2018, Figure 5).
-	The paper is not consistent in notation. The authors introduce MDPs without states but observations, which is simply not the correct definition. They move on and replace observations with states in section 2.2, without every introducing or defining it (states). Lastly, Fig.2 uses X_t, which is also never introduced or defined.
-	In 2.5, the authors communicate their choice for different parameters as based on ‘early experiments’. Without further clarification, this statement is not meaningful as it is highly ambiguous.

Despite the aforementioned issues, the paper is mostly well-written and the presented idea is valid and important. As the main criticism is the choice of random seeds in the experiments, I would accept the paper if this is accordingly addressed.

------------------------------------------------------------------

The authors sufficiently addressed all questions in their rebuttal and I will therefore increase my score.

---

> ### Author Response · Authors · 2020-11-12
> **Response for AnonReviewer4**
>
> Thanks for your comments! We understand your concerns, and we put in a lot of work to be diligent about evaluation ourselves compared to prior work (adding additional robust metrics, re-implementing baselines, plotting the whole distribution of scores, using more random seeds, reducing variance by averaging over more episodes during eval, providing the source code during review, etc.). We hope our clarifications below are convincing of a sound methodology:
> * We should have been more descriptive in the caption about what Figure 3 is plotting.  The boxplot in Figure 3 shows the distribution of scores across different Atari games, not different random seeds. Each point in the boxplot is the score for a single game, averaged across multiple random seeds. The boxplot whiskers are not error bars representing outcomes of different random seeds; they show the interquartile range of human-normalized scores over the set of tested games. We will update our draft to make this clear in the caption.
> * Following Henderson et. al, we move away from the practice of averaging scores over only 5 seeds as done in most previous works, to averaging over 10 seeds in our paper. Even outside the 100k regime, most papers on Atari only report results averaged over 3-5 seeds, so we believe using 10 seeds is an improvement in this aspect. Moreover, following Henderson et. al, we do not do any cherry picking of random seeds, and use the same 10 random seeds for all experiments.  We also recommend the use of multiple metrics in evaluating algorithms that we’ve reported, which can make judgments somewhat more robust to variance.
> * When comparing against prior works, we directly report the scores presented in their papers. However, in order to perform controlled experiments, we re-implemented baseline versions of Rainbow (with and without data augmentation) and ensured all hyper-parameters match with SPR.  All experiments with these baselines are run with the same 10 random seeds as SPR, allowing maximally fair comparisons. We refer to this controlled baseline as “Rainbow (controlled)” in the paper.
> * Thanks for pointing out the inconsistency with observations vs states! This seems to be a remnant of an earlier draft of the paper that used a POMDP-style formulation.  We’ve standardized everything on states for the final version, including changing the figure.
> * We chose lambda=2 by tuning SPR on a small number of games, which is computationally efficient and is also a recommendation in prior literature (Bellemare et. al 2013, Machado et. al 2017). We’ve now clarified this in the main text.

---

> > ### Comment · AnonReviewer4 · 2020-11-18
> > **Response to rebuttal**
> >
> > Dear Authors,
> >
> > Thank you for the clarifications. I appreciate you explaining Figure 3 in more detail and also clarifying the experimental design, with respect to its statistics.
> > I agree that 10 seeds is an improvement over the usual way experiments are performed and reported. The comparison with prior work was a little bit confusing to me (hence my questions in the review) - so please carefully re-read this part.
> >
> > I feel that all my concerns are meaningfully addressed and I will therefore increase my score for the paper.

---

### Official Review · AnonReviewer3 · 2020-10-31
**Comprehensive experiments but can use more analysis**

**Rating:** 7
**Confidence:** 4

**Review:**

This paper proposes Self-Predictive Representations (SPR), a self-supervised representation learning algorithm designed to improve the data efficiency of deep reinforcement learning agents. The main idea is to maintain a target encoder network and a target projection network (the moving average of the corresponding online networks) which can be used as a similarity prediction loss for transitions in the latent space. This loss essentially replaces the reconstruction loss typically used in similar latent space models.

The paper is very well written and easy to read and understand. Figure 2 clearly demonstrates the proposed method and the text does a good job at filling in the details. The experiments are relatively comprehensive (using the 100K sample threshold on Atari games proposed by SimPLe) and the authors compared the performance of the proposed methods with majority of key works on rl from pixels. The appendix also provides more detailed results and the implementation details. The authors also included an anonymous link to a repo which includes the code which is always a plus. The implementation is based on Rainbow as mentioned in the paper.

The main idea of he paper is very intriguing: that the learned state representation can be improved by self forward prediction. The idea is generic enough to be used in similar problems to improve the sota models.  However, I wish the Analysis section was bigger than half a page as I have hard time understanding *where* this improvement is coming from. There are interesting  ablation studies and insights in Section 5 but the authors could provide more. For example, the role of input augmentation, prediction horizon (dynamic modeling in the text), the magic of target network as well as the importance of tau, could be disentangle and explored more.

The paper can also be improved further by including a wall-clock time analysis. The method is clearly outperforming Rainbow but it comes at the cost extra parameters and computation. The authors listed "not relying on reconstructing raw states" as a benefit of the proposed method but I'm curious how does the proposed method compares to such methods, both the ones that unroll in the pixel space (e.g. SimPLe) and the ones that only use it as an auxiliary loss during training (e.g. Dreamer). By comparison I mean in terms of performance and wall clock time. Regardless, Dreamer seems to be a an interesting base model to compare to.

---

> ### Author Response · Authors · 2020-11-12
> **Response to AnonReviewer3**
>
> Thank you for the comments!
> * Due to space constraints, many of the questions you raise are addressed in the appendix rather than the analysis section.  In particular, we address the role of EMA parameter tau and the importance of using a target network with experiments in Appendix C.   We also touch on the role of the prediction horizon in Table 2.   The “Non-temporal SPR” experiment in table 2 can also be understood to show that while SPR may derive some benefit from learning invariance to augmentation (as in BYOL), the key ingredient in SPR is temporal prediction.
> * We’ll include a mention of wall-clock time in the final paper.  In general, however, SPR adds only a small amount of overhead, with 100k steps in Atari taking SPR just over four hours on a V100 GPU, compared to just over three hours for our baseline DQN.  SPR also doesn’t add very many extra parameters -- the transition model’s convolutional layers have at most 85k parameters, and the predictor roughly another 250k (a single 512 x 512 linear layer).   For comparison, the base Rainbow DQN has several million parameters (exact numbers vary by game, as not all games have the same number of actions).
> * As far as we know, Dreamer and DreamerV2 have not been tested in the data-efficient setting on Atari, so we can’t compare to Dreamer directly.  However, we expect that SPR would be somewhat more time-efficient than Dreamer, since SPR does not do reconstruction and thus does not need a deconvolutional network.  SPR also has far fewer parameters than DreamerV2 (less than 5 million for SPR, vs over 20 million).
> * As for SimPLe, SPR is dramatically faster.  SimPLe requires roughly three weeks to run on a single game (see page 5 of https://arxiv.org/pdf/1903.00374v3.pdf), and has roughly 75M parameters in its transition model, which needs to be large because it operates directly at the pixel level.  SPR, on the other hand, adds only about 300k parameters to a regular agent and runs in roughly four hours.  SPR also achieves much stronger performance than SimPLe, as can be seen in the paper.

---

### Official Review · AnonReviewer2 · 2020-11-03
**a data-efficient RL method with strong empirical performance**

**Rating:** 7
**Confidence:** 4

**Review:**

This paper proposes a method to improve data-efficiency of DQN (specially, the Rainbow DQN) by training the reinforcement learning agent to jointly minimize the DQN loss and a loss for multi-step predictions on its own latent states.

Pros
+ The proposed approach achieves signficant performance improvement as compared to existing algorithms in the low data regime on Atari games.
+ The paper is well-written and generally easy to follow. Fig. 2 provides a nice illustration on the algorithm.

Cons
The approach provides an interesting combination of several heuristics, but it isn't always clear how these work and whether these are necessary components.
- Both the target encoder network and the target projection head uses the EMA trick, and an additional prediction head q is used. There is some discussion on the EMA trick in the experiment section for the target encoder, but there is little discussion on the the target projection head and the prediction head.
- Why is the prediction loss chosen to be negative cosine similarity? Does using quadratic loss work?

Minor comments
- Alg. 1: a loop over experiences in the minibatch seems to be missing, and RL loss(s, a, r, s'; \theta_{o}) should be a function of the mini-batch not just a single experience (s, a, r, s'). Is s' perturbed as well?
- Specifying the network parameters in notations will improve clarity (though many papers don't). In this paper, sometimes the network parameters are included in a notation, but sometimes not (e.g. L^{RL} and $L^{RL}_{\theta}).
- Strictly speaking, cosine similarity = 1 - 0.5 * normalized l2 loss (not really proportional)
- Is the algorithm's performance sensitive to the choice of K?
- The paper states that "using a separate target encoder is vital in all cases", this is confusing as the paper states later that with data augmentation, this is not needed (\tau=0 works best).

---

> ### Author Response · Authors · 2020-11-12
> **Response to AnonReviewer2**
>
> Thank you, we spent a lot of time ensuring the paper communicates the core ideas well both via text and illustrations, so we are glad that’s reflected in the final version. In regards to questions and comments, here’s our brief response:
> * Intuitively, using a projection head serves two purposes.  First, by defining the projection to be the first layer of the DQN head, we calculate the SPR loss on the rich representations used to predict value, rather than the convolutional features, which are much higher-dimensional and may contain extraneous information.  Second, this parameter sharing serves to increase the impact of the SPR objective by allowing it to affect more parameters; the linear layer used as projection head has over 1.5M parameters, compared to under 100k in the convolutional encoder.  Using this type of projection has become quite common in self-supervised learning tasks (see SimCLR and BYOL), and we found it beneficial in SPR.
> When using a projection head and an EMA target encoder, it is logical to use a matching EMA target projection head to generate target representations.  If the online projection head were used without a stopgradient, the projection head would be encouraged by the optimization process to collapse all the targets and predictions to a single point, which would be undesirable (see experiments in Table 7, which test a case where gradients are allowed to propagate through the SPR targets).  A detached version of the online projection head could be used, but this might suffer from distributional shift, as the target encoder’s outputs may be systematically different from the outputs of the online encoder.
> Following BYOL, we also include a prediction head, which translates outputs from the online projection head to match those of the target projection head.  As the predictor is applied only to online representations, no EMA copy is needed.
>
> * Maximizing cosine similarity has been shown to be quite an effective objective in representation learning. In particular, the objective used in recent contrastive works such as SimCLR can be understood as maximizing the cosine similarity between representations of positive pairs while minimizing it between negative pairs, while BYOL directly maximizes cosine similarity between positive pairs without reference to negative pairs.
>
>  In SPR, maximizing cosine similarity has the advantage of making the scale of the SPR loss independent of the scale of the representations.  If we were to minimize a quadratic error, the network could learn to “cheat” by making its activation norms smaller; if activations were shrunk by a constant multiple C and the weights in the final output layer multiplied by the same C, the SPR loss could be made arbitrarily small without affecting the reinforcement learning loss at all. On the flip side, if the network’s activations were to become large, the SPR loss might grow to the point of causing instability in optimization, and entirely outweigh the reinforcement learning loss.
>
> With cosine similarity, the loss has a strict cap, so divergence is less of an issue.  Moreover, activation norm collapse does not reduce the SPR loss, removing one spurious minimum in the optimization process.
>
> *Minor Comments.*  Thanks for pointing these out.
> * We’ll take a look at refactoring the pseudocode to make things more obvious.  $S’$ is also perturbed, when augmentation is in use.
> * We’ll standardize notations on $_{\theta}$
> * That’s true, we were thinking of the gradients.  We’ll clarify this.
> * We find that longer prediction horizons (K) improve performance, in particular in one of our ablations, we compare K=1 and K=5. We chose 5 because it matches the value chosen in MuZero; increasing the prediction horizon further might provide further marginal improvements in performance, but would come at a computational cost.
> * By separate target encoder in this case, we mean a gradient-detached version of the online encoder.  Using the online encoder directly results in dramatic reductions in performance (see the ablations in appendix C).

---

### Author Response · Authors · 2020-11-19
**Revision with suggested changes**

We've updated our manuscript following some great feedback and comments from the reviewers. Here's a brief changelog of the the changes we added:
* A new ablation sweeping over prediction horizons from K=0 to 8. We find that increasing the prediction horizon almost monotonically leads to increase in performance, with improvements starting to taper off after K=5.
* A new ablation examining the effect of using a normalized L2 loss vs using an un-normalized L2 loss. As suspected, the un-normalized L2 loss collapses as previously observed in Gelada et al. 2017 (section C.5) [1].
* A new ablation examining the effect of using projection heads, re-affirming our design choice that using projection heads like prior work leads to improved performance.
* A comparison of wall-clock running times for a subset of methods discussed in the paper. SPR is able to run in just 4.5 hours for a single game on a P100 GPU, which is quite efficient compared to over 500 hours of a pixel-reconstruction method like SimPLE.
* Improved clarity on what the boxplot in Figure 3 is plotting.
* Fixed notation issues around the MDP definition and the algorithm box.
* Future work section now includes a paragraph on investigating whether SPR can help RL algorithms to generalize better to unseen tasks or environments.

[1] Gelada, Carles, et al. "Deepmdp: Learning continuous latent space models for representation learning." arXiv preprint arXiv:1906.02736 (2019).

---

### Decision · Program_Chairs · 2021-01-07
**Final Decision**

**Decision:**

Accept (Spotlight)

**Comment:**

The authors propose self predictive representations (predicting the agents own future latents of a forward model with data augmentation) as a means of improving the data efficiency of agents. The reviewers found the paper to be compelling (after the authors made adjustments) and pointed out that the method is likely generic and might be widely applicable. Experimental improvements in the work are significant, and the method is well explored.